# Microbiome-Based Therapeutics for Insomnia

**DOI:** 10.3390/ijms252313208

**Published:** 2024-12-09

**Authors:** Chenyu Li, Sizhe Chen, Yun Wang, Qi Su

**Affiliations:** 1Microbiota I-Center (MagIC), Hong Kong SAR, China; 2Department of Medicine and Therapeutics, The Chinese University of Hong Kong, Hong Kong SAR, China

**Keywords:** insomnia, sleep quality, short-chain fatty acids, dysbiosis, probiotics, fecal microbiota transplantation

## Abstract

Insomnia poses considerable risks to both physical and mental health, leading to cognitive impairment, weakened immune function, metabolic dysfunction, cardiovascular issues, and reduced quality of life. Given the significant global increase in insomnia and the growing scientific evidence connecting gut microbiota to this disorder, targeting gut microbiota as an intervention for insomnia has gained popularity. In this review, we summarize current microbiome-based therapeutics for insomnia, including dietary modifications; probiotic, prebiotic, postbiotic, and synbiotic interventions; and fecal microbiota transplantation. Moreover, we assess the capabilities and weaknesses of these technologies to offer valuable insights for future studies.

## 1. Insomnia: An Academic Perspective

Insomnia is a common sleep disorder characterized by difficulties in initiating sleep or maintaining sleeping state, along with daytime impairment symptoms [1]. Nearly 40% of adults present insomnia symptoms and 10.8% of adults suffer from chronic insomnia disorder [2]. Particularly, the elderly patients with chronic diseases and individuals with high stress are highly vulnerable to insomnia [3]. Sleep loss raises the risks of chronic diseases (e.g., cardiovascular disease, diabetes, and obesity) and psychological disorders (e.g., depression and anxiety) [4,5,6,7]. Commonly employed assessment strategies for insomnia include self-report questionnaires (e.g., Pittsburgh Sleep Quality Index (PSQI), Oguri–Shirakawa–Azumi sleep inventory, and Insomnia Severity Index) and polysomnography monitoring (e.g., electroencephalography (EEG), eye movements, and electromyography), which measures sleep in terms of sleep status, insomnia symptoms, and sleep quality [8,9]. Insomnia treatment aims to enhance both quality and quantity of sleep [10]. Current insomnia therapies mainly include medications and psychological therapies. However, medications like benzodiazepine have adverse effects, such as excessive neurological toxicity, addiction, and tolerance [11]. Psychological therapies, such as cognitive behavioral therapy, have demonstrated long-term effectiveness. However, their availability is limited due to a shortage of trained therapists and high costs [12]. With the development of sequencing and multi-omics analysis technologies, current understanding of microbiota biology has revealed the significant role of gut microbiota in regulating sleep, highlighting the potential of microbiome-based therapeutics for alleviating insomnia symptoms [13].

## 2. The Significant Role of Gut Microbiota in Regulating Sleep

The human gut intrinsically harbors large quantities of microbes, including bacteria, fungi, viruses, and archaea [14]. Gut microbiota plays a crucial role in maintaining homeostasis by participating in metabolic processes, supporting immune function, and influencing neurological health [15]. The composition of gut microbiota can be altered by host-specific factors such as genetic background, age, and health status, and environmental factors like diet, exercise, xenobiotics (e.g., antibiotics, prebiotics, probiotics, and postbiotics), and hygiene [16]. An imbalance in gut microbiota composition, known as dysbiosis, significantly impacts sleep efficiency. For example, mice with depleted microbiota exhibited a significant reduction in non-rapid eye movement (NREM) sleep during their inactive phase [17]. Conversely, during the active phase, these mice spent more time in both NREM sleep and rapid eye movement (REM) sleep, along with an increased frequency of transitions between NREM and REM sleep. Clinical studies have indicated that there are distinct differences in the composition, diversity, and metabolic functions of gut microbiota between healthy individuals and those with insomnia [18]. Insomniacs exhibit low microbial diversity, a decreased abundance of short-chain fatty acid (SCFA)-producing bacteria, and an expansion of potential pathobionts [19]. At the phylum level, insomniacs show higher relative abundance of *Bacteroidetes* and lower relative abundance of *Firmicutes* and *Actinobacteria* than healthy individuals [18,20,21]. At the genus level, genus *Gemmiger* and *Fusicatenibacter* are dominant bacterial genus in patients with insomnia disorder [18]. Given the emerging correlation between gut microbiota and sleep, strategies aiming to restore microbial balance offer new avenues for treating insomnia. In this review, we summarize the advanced microbiome-based therapies for insomnia, including dietary modifications; probiotic, prebiotic, postbiotic, and synbiotic interventions; and fecal microbiota transplantation (FMT), indicating the promising capability of insomnia therapeutics by delicately mediating gut microbiome profiles.

## 3. Potential Insomnia-Relevant Pathways Regulated by Gut Microbiota

As shown in Figure 1, the basis for microbiome-based therapeutics for insomnia lies in the relationship between gut microbiota and the brain. The gut microbiota influences sleep through the brain–gut microbiota axis, a bidirectional communication system that consists of the hypothalamic–pituitary–adrenal axis, microbiota metabolites, the immune system, and the vagus nerve.

### 3.1. Gut Microbiota Regulates the HPA Axis to Affect Sleep

The HPA axis is the neuroendocrine system responsible for coordinating the body’s response to stress by regulating hormones such as adrenocorticotropic hormone (ACTH) and cortisol [22]. During sleep onset, HPA axis activity is suppressed, leading to a reduction in the secretion of ACTH and cortisol. Conversely, upon awakening, HPA axis activity increases, resulting in heightened plasma levels of ACTH and cortisol [23,24]. There is a bidirectional relationship between the HPA axis and insomnia: an activated HPA axis can lead to lighter sleep and more frequent nighttime awakenings, while insomniacs tend to have significantly higher 24 h plasma levels of ACTH and cortisol compared to healthy individuals [25]. The gut microbiota can stimulate the HPA axis through microbial antigens, cytokines, and prostaglandins, while its metabolites, SCFAs, can attenuate the response of the HPA axis [26]. Therefore, regulating HPA axis activity through the gut microbiota is a potential approach for treating insomnia.

### 3.2. Microbiota-Derived Metabolites and Their Impact on Sleep

The primary metabolites of the gut microbiota, such as SCFAs and neurotransmitters, play vital roles in regulating sleep.

#### 3.2.1. SCFAs

SCFAs are the end products of anaerobic bacterial fermentation in the gastrointestinal tract, primarily comprising acetate, propionate, and butyrate [27]. Research indicates that the relative abundance of SCFA-producing bacteria is lower in patients with insomnia [19]. SCFAs regulate sleep by enhancing neurotransmitter production, exerting anti-inflammatory effects, and facilitating circadian rhythm adjustments [28,29,30]. In the gut, SCFAs can promote serotonin secretion by boosting the levels of tryptophan hydroxylase 1 or by binding to specific receptors on enterochromaffin cells [28]. Additionally, SCFAs can suppress the production of pro-inflammatory cytokines while promoting the release of anti-inflammatory cytokines and T cell differentiation [29]. These actions help alleviate neuroinflammation in the brain and improve sleep quality. SCFAs have been shown to influence the expression of circadian clock genes, aiding in the adjustment of circadian rhythms [30]. Notably, a recent study in rodents demonstrated that the administration of butyrate resulted in a nearly 50% increase in NREM sleep in mice [31].

#### 3.2.2. Neurotransmitter

The gut microbiota is capable of producing various neurotransmitters, including gamma-aminobutyric acid (GABA) and serotonin, both of which play significant roles in sleep regulation. GABA is the main inhibitory neurotransmitter of the central nervous system and the activation of GABA receptors promotes sleep [32]. GABA primarily inhibits the activity of glutamatergic neurons and their associated receptors, thereby promoting NREM sleep [33]. Patients with primary insomnia exhibited approximately 30% reduction in brain GABA levels compared to healthy subjects [34]. The transcriptome analysis of stool samples from healthy individuals revealed that GABA-producing pathways are actively expressed in *Bacteroides*, *Parabacteroides*, and *Escherichia* species [35].

Serotonin is a monoamine transmitter that mainly exists in the cerebral cortex and synapses [36]. The serotonergic system participates in sleep–wake behavior and sleep architecture. Rats with serotonin deficiency in the brain displayed disrupted circadian rhythms in their sleep–wake cycles, characterized by reduced NREM sleep during the inactive phase and increased slow-wave sleep duration during the active phase [37]. Mice with depleted serotonin induced by broad-spectrum antibiotics exhibited less NREM sleep during the inactive phase as well [17]. Para-chlorophenylalanine (PCPA) can exhaust serotonin in the brain and induce insomnia, but sleep behavior could be recovered by the administration of a serotonin precursor [38]. Certain prebiotics and probiotics can elevate the levels of the serotonin precursor tryptophan by modulating its production [39]. Indigenous spore-forming bacteria from mice and humans can promote serotonin synthesis in the intestine [40]. Serotonin is also the precursor to the hormone melatonin, which regulates circadian rhythms and facilitates sleep [41]. Diminished melatonin secretion has been observed in elderly insomnia patients [42]. Melatonin supplementation has been reported to improve sleep efficiency and shorten sleep latency [43,44].

### 3.3. Gut Microbiota Participates in Stimulating the Immune System Induced by Sleep Disruption

Chronic insomnia can stimulate the immune system, activate inflammatory responses, and increase oxidative stress, with gut microbiota contributing significantly to these processes [45]. Studies revealed that insomniacs exhibited increased IL-1β levels and reduced TNF-α levels. The change in immune factors is associated with a higher relative abundance of *Prevotella*, which is closely associated with various immune-related diseases [46]. SCFAs produced by gut microbiota can inhibit the release of inflammatory cytokines such as TNF-α and IL-6 [47]. Probiotic supplementation attenuates inflammation and oxidative stress in mice by boosting the antioxidant capacity of their brains [48].

### 3.4. Gut Microbiota Activate Vagus Nerve to Affect Sleep

The vagus nerve mediates the connections among the gut microbiota, the brain, and sleep. The ingestion of *Lactobacillus rhamnosus* altered the expression of central GABA receptors and reduced anxiety and depression-related behaviors in normal mice, but not in vagotomized mice [49]. Additionally, transferring stool microbiota from mice with systemic inflammation caused by sleep deprivation resulted in elevated plasma levels of inflammatory markers in germ-free (GF) mice, but this effect was not observed in vagotomized GF mice [50].

## 4. Microbiome-Based Therapeutics for Insomnia

In this section, we focus on microbiota-based therapeutics for insomnia, including dietary modifications; prebiotic, probiotic, postbiotic, and symbiotic interventions; and FMT. As shown in Figure 2, all of them offer valuable insights for improving sleep.

### 4.1. Diet Modification

Dietary patterns are defined as the quantities, proportions, variety, or combinations of different foods, drinks, and nutrients in diets [51]. After reaching adulthood, individuals primarily depend on healthy diets to maintain a beneficial microbial composition, which plays a vital role in regulating sleep. Global metabolic profiles indicate that insomnia patients exhibit significantly higher levels of energy metabolites compared to healthy controls [52]. Additionally, obese individuals show a notable decrease in the abundance of the *Faecalibacterium*, which is also observed in patients with chronic insomnia [19,53]. However, a calorie-restricted diet combined with increased physical activity can help obese adolescents increase the abundance of the *Lactobacillus* group in their gut, which is known for its beneficial role in promoting sleep [54]. A high glycemic index and glycemic load diet are associated with an increased risk of insomnia and poor sleep quality [55,56]. This is because prolonged high sugar intake can stimulate the immune system and induce excessive inflammatory responses, leading to fragmented sleep [57,58]. Conversely, the Mediterranean diet—a dietary pattern rich in plants, antioxidants, and unsaturated fats—has been linked to improved sleep quality and longer sleep duration [59]. The potential reasons lie in the induced higher abundance of butyrate-producing bacteria and significant changes in the metabolic activity of the gut microbiota [60]. The sleep-improving effects of some functional foods have also been reported. Research indicates that Ganoderma lucidum can reduce sleep latency and prolong sleep duration in mice by altering the gut microbiota composition with enriched beneficial bacteria and their metabolites, including *Bifidobacterium*, indole-3-carboxylic acid, and acetyl phosphate [61]. Furthermore, habitual tea consumption can alleviate the imbalance of gut microbiota and metabolic disorder of bile acid caused by chronic insomnia through the gut microbiota-bile acid axis [62].

### 4.2. Prebiotics

Prebiotics are nondigestible food ingredients that benefit host health by selectively stimulating the growth and activity of specific bacteria in the colon. Rodent studies have demonstrated the positive effects of prebiotics on sleep regulation. Prebiotic diet (composed of galactooligosaccharides, polydextrose, lactoferrin, and whey protein concentrate milk fat globular membrane protein-10) positively influenced sleep patterns of mice by enhancing NREM sleep in normal conditions and prolonging REM sleep during stressful situations [63]. Fecal metabolism analysis showed a prebiotic diet reduced allotetrahydrodeoxycorticosterone levels, which metabolite can inhibit the GABAergic system and has also been associated with poor sleep quality during pregnancy. Another study showed that rats with a prebiotic diet (composed of galactooligosaccharides and polydextrose) showed longer NREM and REM sleep under sleep disruption compared to rats with a control diet [64]. The microbiome analysis found an increase in the relative abundance of the species *Parabacteroides distasonis* in the prebiotic group, which is associated with facilitated sleep and recovered circadian clock [65]. Rats treated with a diet containing prebiotics exhibited better sleep efficiency when exposed to stress, and higher levels of alpha diversity and increased relative abundance of *Lactobacillus rhamnosus* were found in their feces [66].

The beneficial effects of prebiotics have also been demonstrated in clinical trials. In a human study, 45 participants were randomly divided into three groups, receiving Fructooligosaccharide (FOS), galactooligosaccharide (GOS), or a placebo, respectively [67]. The results indicated that participants in the GOS group experienced improved sleep quality and a significantly lower salivary cortisol awakening response. This reduction in cortisol was correlated with a higher relative abundance of *Bifidobacteria* in the gut microbiota, which are recognized for their ability to metabolize GOS and contribute positively to mental health [68]. Resistant dextrin is another prebiotic that has been shown to improve sleep quality in female patients with type 2 diabetes by reducing inflammation and HPA axis activity [69]. A randomized, double blind, and placebo controlled study also reported the sleep-improving effects of prebiotic yeast mannan [70]. EEG monitoring results indicated that the duration of NREM stage 3 sleep and total bedtime in the yeast mannan group were significantly greater than in the placebo group. Additionally, metabolomic analysis identified alterations in fecal propionate and GABA levels as key factors contributing to the observed sleep-enhancing effects of the prebiotic intervention.

Many traditional Chinese medicine prescription act similarly to prebiotics and exert sleep-promoting effects. Jiaotai pills composed of Coptidis rhizome and Cinnamomi cortex are reported to enhance sleep and attenuate the inflammation and insulin resistance induced by chronic sleep deprivation [71]. Omics results showed Jiaotai pill treatment significantly increased the relative abundance of *Lachnospiraceae*, *Bacteroides*, and *Akkermansia* in mice. Bailemian can significantly alleviate PCPA-induced insomnia in mice by remodeling the composition of the gut microbiota and recovering neurotransmitter levels [72].

### 4.3. Probiotics

Probiotics are live microorganisms that support host health by promoting a balanced gut microbiota composition [73]. Probiotics can remodel the gut microbiota by competing for nutrients with harmful bacteria, producing beneficial metabolites, and modulating the host’s immune response [74]. With the growing understanding of the microbiome’s role in sleep, probiotic supplementation has emerged as an effective therapy for alleviating insomnia (Table 1). Rodent studies have demonstrated the efficacy of probiotics supplement in sleep improvement. Mice administered milk fermented with the probiotic *Lactobacillus brevis* DL1-11 exhibited shorter sleep latency and longer sleep duration compared to those receiving a placebo [75]. Because DL1-11 can produce large amounts of GABA, which can promote relaxation and enhance sleep. Another study reported that *Lactobacillus fermentum* PS150^TM^ effectively promoted sleep in normal mice and alleviated insomnia symptoms in caffeine-induced insomnia mice by increasing the expression of adenosine A1 receptors in the hypothalamus [76]. The same team also discovered that PS150^TM^ could restore NREM sleep in mice with sleep disturbance caused by the first night effect (FNE), likely due to its ability to remodel the gut microbiota composition [77]. The FNE refers to a phenomenon observed in sleep studies where individuals exhibit heightened alertness and anxiety when exposed to an unfamiliar environment, often resulting in acute insomnia [78]. There is no doubt that sleep disorders are closely linked to levels of stress and anxiety. Moreover, many probiotics have been shown to concurrently improve both sleep quality and mental health. PS150^TM^ has been shown to reduce depression and anxiety in rats with chronic mild stress by altering the serotonergic pathway [79]. This highlights the potential of psychobiotics—probiotics yielding mental health benefits when administered in adequate amounts—in enhancing sleep [80].

Numerous human trials have demonstrated the promise of psychobiotics in enhancing sleep. *Bifidobacterium* supplementation has garnered attention for its potential to improve sleep quality, particularly in individuals experiencing stress. *Bifidobacterium breve* CCFM1025 improved sleep and alleviated stress levels in patients diagnosed with stress-induced insomnia by suppressing HPA axis activity [81]. *Bifidobacterium longum* 1714 treatment significantly improved sleep quality and reduced daytime dysfunction in healthy adults experiencing sleep impairment due to examination stress [82]. However, the EEG monitoring results in this study did not exhibit significant changes. Complementary research on 1714 indicated no significant change in PSQI scores but noted an increase in overall sleep duration [83]. *Bifidobacterium adolescentis* SBT2786 could prolong total sleep duration by increasing REM sleep time, though it did not enhance subjective sleep quality [84]. Interestingly, researchers found that participants with high stress levels experienced greater benefits from SBT2786, showing significant improvements in sleep duration and reduced daytime fatigue. Although there is insufficient evidence to classify SBT2786 as a psychobiotic, the subgroup analysis of individuals under high stress indicates its potential for mental health.

Many *Lactobacillus* strains are also potential psychobiotics that can promote sleep. *Lactobacillus plantarum* PS128 significantly enhanced deep sleep and decreased fatigue and depressive symptoms in self-reported insomniacs [85]. A rodent study showed PS128 could increase serotonin and dopamine in the striatum, which may be the underlying mechanism of the beneficial effects [86]. *Lactobacillus gasseri* CP2305 has also been shown to improve intestinal microbiota composition and reduce salivary cortisol concentrations, thereby alleviating stress and enhancing sleep [87]. *Lactobacillus casei* Shirota milk helped individuals facing academic examination stress by prolonging sleep duration and reducing feelings of sleepiness upon awakening [88]. EEG monitoring further revealed the interactions among sleep, stress, and probiotics. As the exam approached, participants experienced increased difficulties with sleep, including prolonged sleep latency and reduced NREM stage 3 sleep duration. However, the intervention with Shirota significantly mitigated these adverse effects. Shirota has also been reported to effectively improve daytime performance in individuals experiencing sleep disorders [89]. *Lactococcus lactis* subsp. *cremoris* YRC3780 also demonstrates potential for improving sleep and mitigating stress, as evidenced by a double-blind, placebo-controlled clinical trial conducted in Japan [90]. The study reported significant improvements in subjective sleep quality and mental health among participants receiving the YRC3780 intervention. Although no substantial differences were observed in EEG recordings, lower morning cortisol levels were noted in the YRC3780 group. This suggests that the objective enhancement in sleep may be associated with a reduced stress response within the HPA axis.

Probiotics not only improve sleep quality but also mitigate the detrimental effects associated with sleep loss. Pre-colonization with *Faecalibacterium prausnitzii* can inhibit inflammatory responses, reduce apoptosis, and protect intestinal barrier integrity in mice with sleep deprivation [91]. The underlying mechanism is that *F. prausnitzii* restores the gut microbiota homeostasis by reducing the abundance of harmful bacteria, such as *Klebsiella* and *Staphylococcus*, while simultaneously promoting the growth of beneficial bacteria, such as *Akkermansia.* Pre-colonization with *Lactiplantibacillus plantarum* 124 also alleviated oxidative stress, inflammation, colonic barrier damage, and the dysbiosis induced by sleep deprivation [92]. Gut microbiota analysis showed that 124 balanced the gut microbiota by restoring the abundances of beneficial bacteria such as *Dubosiella*, *Faecalibaculum*, *Bacillus*, *Lachnoclostridium*, and *Bifidobacterium.*

### 4.4. Postbiotics

The definition of postbiotics is non-living microorganisms and their components that provide health benefits to the host [93]. They encompass inactivated probiotics along with their metabolic products like SCFAs, vitamins, organic acids, and cellular components such as teichoic acids and peptidoglycan, which still confer health benefits to the host in a non-living state [94]. Compared with probiotics, postbiotics are safer as they have no biological activity. Rodent studies have confirmed that SCFAs can prolong sleep, as evidenced by direct injections of butyrate in mice [31]. Additionally, human studies have indicated that treatment with a mixture of SCFAs significantly attenuated the cortisol response, which is beneficial for sleep onset [95]. Lipoteichoic acid, a cell wall component of Gram-positive bacteria, has been found to dose-dependently increase NREM sleep in mice [96]. Dietary supplementation with heat-killed Lactobacillus brevis SBC8803 significantly decreased NREM sleep during the active phase and increased NREM sleep time during the inactive phase in mice [97]. SBC8803 has been reported to regulate vagus nerve activity and promote serotonin release [98,99]. Furthermore, heat-killed Levilactobacillus brevis increased NREM and REM sleep of mice during their inactive phase, and heat-killed *Lactobacillus gasseri* CP2305 improved sleep disturbance in chronically stressed individuals [100].

### 4.5. Synbiotics

Synbiotics are mixtures of probiotics and prebiotics that work synergistically to enhance host health. Although there are few studies specifically examining the effects of synbiotics on pure insomnia, research indicates its promising outcomes. The results of a double-blind, randomized controlled trial showed that a synbiotic preparation (SIM01) effectively alleviated post-acute COVID-19 insomnia [101]. Additionally, another clinical study suggested that synbiotic supplementation could improve the sleep quality of patients in end-stage renal disease undergoing hemodialysis [102].

### 4.6. FMT

FMT is a medical procedure that involves transferring stool from a healthy donor into the gastrointestinal tract of a patient [103]. A diverse range of microorganisms from a healthy donor can help repopulate the gut with beneficial bacteria, and the restored balance of gut microbiota works as a host defense against harmful pathogens. FMT has shown promising results in the treatment of ulcerative colitis associated with inflammatory bowel disease (IBD) [104]. Currently, an increasing number of studies are focusing on FMT as a potential treatment for mental health disorders, given that the gut microbiota can impact the central nervous system via the microbiota–brain axis (Table 2). A clinical study has demonstrated that washed microbiota transplantation (WMT) significantly improves sleep disorders in patients with various underlying conditions [105]. The 16s rRNA results showed that the WMT increased the relative abundance of beneficial bacteria, such as *Bifidobacterium*, *Prevotella 7*, *Ruminococcus gnavus*, and *Faecalibacterium*, and reduced the relative abundance of harmful bacteria, such as *Escherichia-Shigella* and *Streptococcus.* WMT also assisted patients with irritable bowel syndrome (IBS) who experience sleep disturbances and psychological disorders by improving sleep efficiency, enhancing mental health, and alleviating gastrointestinal symptoms [106]. Notably, patients who experienced improvements in sleep quality also showed greater reductions in depression and the severity of IBS symptoms. This suggests that the restructured gut microbiota leads to comprehensive improvements for patients. Another open-label observational study found similar results in patients with gastrointestinal diseases experiencing sleep disturbances, and the authors suggested that the overall improvement was attributed to the increased microbiota diversity following FMT treatment [107]. A real-world study further connected the link among FMT, probiotics, and chronic insomnia [108]. Patients with various chronic disorders were randomly divided into two groups based on the presence of insomnia and received a 4-week FMT treatment. The results indicated that FMT significantly improved sleep quality, reduced sleep latency, and increased sleep efficiency in patients experiencing insomnia. By comparing gut microbiota composition, researchers found that FMT significantly increased the relative abundance of SCFA-producing probiotics, including *Lactobacillus*, *Bifidobacterium*, and *Turicibacter.* However, all these studies investigated insomnia as a comorbidity associated with other diseases, suggesting that improvements in sleep may also be attributed to the alleviation of disease symptoms, and the effectiveness of FMT for sleep disorders requires further investigation. Additionally, the safety of FMT must also be considered [109]. The safety assessments for FMT samples are based on identifying known pathogens and the donors complete health. And due to the limited understanding of the gut microbiota, it is difficult to determine whether the donor’s microbiota may include opportunistic harmful bacteria.

## 5. Conclusions and Prospects

This review describes current microbiome-based therapies for alleviating insomnia symptoms and enhancing sleep quality. Diet shapes the gut microbiota and plays a crucial role in regulating sleep. Individuals with poor dietary habits, such as high energy and sugar consumption, are more prone to suffering insomnia. To improve sleep quality, the Mediterranean diet, prebiotic supplements, and functional foods are highly recommended. Probiotic treatments involving *Bifidobacterium* and *Lactobacillus* have shown promise in improving sleep quality and efficiency, especially for individuals experiencing acute insomnia induced by stress. Probiotic treatments like pre-colonization with specific strains such as *Faecalibacterium* and *Lactiplantibacillus* can significantly mitigate the negative effects of sleep deprivation. Currently there is limited research on the effects of synbiotics for treating pure insomnia, but its potential synergistic impact merits further investigation. Insomnia frequently coexists with other health conditions, and FMT therapy has shown benefits not only in alleviating disease symptoms but also in improving sleep quality.

However, several issues require further exploration:(1)The dosage of probiotics and FMT: Unlike other microbiome-based therapeutics, probiotics, synbiotics, and FMT involve the transfer of live microorganisms into the gut. The efficacy of this transfer does not solely depend on increasing specific probiotic strains; rather, it is more about achieving a balanced microbiota composition that may involve complex synergistic or antagonistic relationships among various microbial species. Numerous studies have shown that higher doses of probiotics do not necessarily yield better outcomes than lower doses, and combinations of multiple strains often demonstrate greater therapeutic efficacy [110]. An insufficient quantity of beneficial microbiomes may fail to remodel gut microbiota composition, while an excess could lead to dysbiosis. Therefore, exploring the appropriate dosage of live microorganisms in treatment is crucial for enhancing therapeutic effects.(2)Dysbiosis and sleep disorders: While the bidirectional relationship between the gut microbiota and sleep is widely acknowledged, a limited number of studies have explored how dysbiosis itself leads to sleep disorders. Investigating how dysbiosis contributes to insomnia and identifying specific microbiome species closely associated with sleep disturbances is important for developing targeted microbiome therapies for alleviating insomnia.(3)Measurement methods: Most clinical studies rely on self-evaluation questionnaires to assess improvements in sleep quality rather than using objective measures like EEG. The lack of objective data complicates the accurate assessment of the efficacy of microbiome-based therapies. Moreover, inconsistent outcomes may negatively impact patients’ choices regarding insomnia treatment.

## Figures and Tables

**Figure 1 ijms-25-13208-f001:**
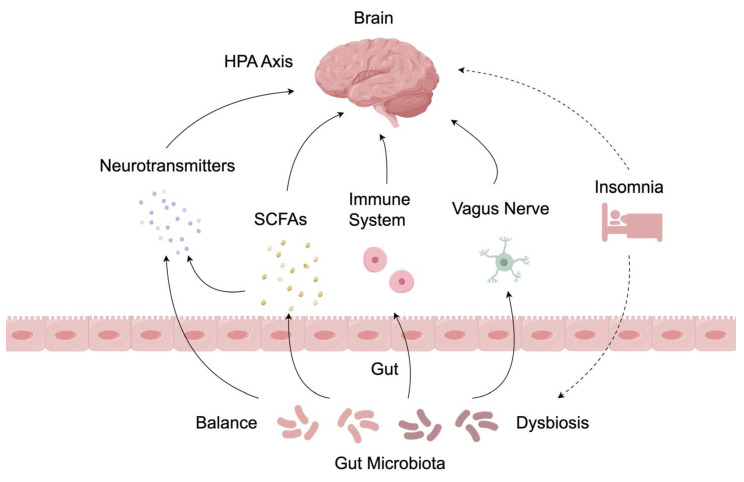
The mechanisms by which the gut microbiota regulates sleep.

**Figure 2 ijms-25-13208-f002:**
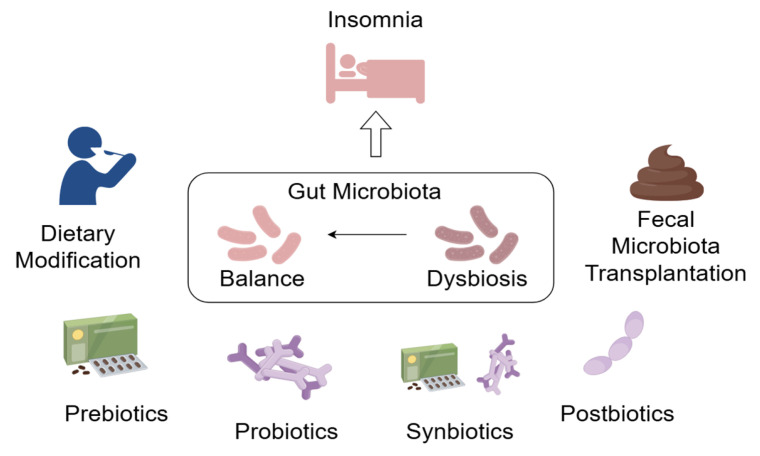
Overview of the current microbiome-based therapeutics for insomnia, including dietary modifications, prebiotics, probiotics, synbiotics, postbiotics, and fecal microbiota transplantation.

**Table 1 ijms-25-13208-t001:** Effects of Probiotics Treatment on Sleep.

Study Model	Treatment	Effect	Ref.
Healthy mice;	Probiotics; *Lactobacillus brevis* DL1-11 milk	Prolonged sleep duration and shortened sleep latency	[75]
Healthy mice; Insomnia mice; rats in novel environment	Psychobiotic; *Lactobacillus fermentum* PS150^TM^	Enhanced sleep in normal mice; alleviated insomnia symptoms in caffeine induced insomnia mice; reduced depression and anxiety-like behaviors and ameliorated sleep disturbance in rats exposed to novel environment	[76,77,79]
Adults with diagnosed insomnia	Psychobiotics; *Bifidobacterium breve* CCFM1025	Improved sleep quality and reduced stress	[81]
Adults with stress- induced sleep impairment	Psychobiotics; *Bifidobacterium longum* 1714	Improved sleep quality and social function, reduced daytime dysfunction, but no significant effect on actigraphy; another study showed 1714 increased sleep duration	[82,83]
Adults dissatisfied with their sleep quality	Probiotics; *Bifidobacterium adolescentis* SBT2786	Increased light sleep duration, improved mood; increased sleep duration and reduced sleepiness upon waking up in individuals with high stress	[84]
Adults with self reported insomnia; GF mice	Psychobiotics; *Lactobacillus plantarum* PS128	Enhanced deep sleep, decreased fatigue and depressive symptoms in self-reported insomniacs; reduced anxiety-like behaviors in GF mice	[85,86]
Healthy men	Psychobiotics; *Lactobacillus gasseri* CP2305	Reduced stress and enhanced sleep	[87]
Healthy adults	Psychobiotics; *Lactobacillus casei* Shirota	Reduced sleepiness upon awaking and prolonged sleep duration	[88,89]
Healthy men	Probiotics; *Lactococcus lactis* subsp. *cremoris* YRC3780	Improved sleep quality and mental health	[90]
Mice with sleep deprivation	Probiotics; *Faecalibacterium prausnitzii*	Reduced inflammatory responses and apoptosis; enhanced intestinal barrier integrity	[91]
Mice with sleep deprivation	Probiotics; *Lactiplantibacillus plantarum* 124	Reduced oxidative stress, inflammation, and colonic barrier damage	[92]

**Table 2 ijms-25-13208-t002:** Effects of FMT treatment on insomnia.

Study Model	Treatment	Effect	Ref.
Patients of various diseases with insomnia	WMT	Improved sleep quality, prolonged sleep duration, and reduced sleep latency	[105]
Patients of IBS with insomnia and psychological disorders	WMT	Improved subjective sleep quality, prolonged sleep duration, and alleviated anxiety, depression, and IBS severity	[106]
Patients of gastrointestinal diseases with insomnia	FMT	Improved sleep, reduced depression, anxiety, and gastrointestinal symptoms	[107]
Patients of various chronic diseases with insomnia	FMT	Improved sleep quality, reduced sleep latency, and increased sleep efficiency	[108]

## Data Availability

Not applicable.

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
