# Peer review of "Microbiome-Based Therapeutics for Insomnia"

_ijms, 2024, doi:10.3390/ijms252313208_

Round 1

Reviewer 1 Report

Comments and Suggestions for Authors

I have read with great interest the manuscript of Li et al evaluating the available literature on Microbiome-based Therapeutics for Insomnia. The subject is of great interest to the scientific community and the manuscript nicely summarizes what is currently known on this matter. The manuscript is very well written, and the title reflects the content of the manuscript.

I have just few minor comments listed below:

1.       For bacteria please italicize family, genus, species, and variety or subspecies, when used in the singular through the manuscript (example, Bacteroides, Parabacteroides, and Escherichia on page 3 lines 126, 127; Lactobacillus rhamnosus page 4 line 154).

2.       Please define all abbreviation at their first occurrence in the text (example, HPA on page 2, line 72; FOS, GOS page 5, line 207)

3.       After an abbreviation was defined use only the abbreviation in the text (example, Fecal Microbiota Transplantation on page 8, line 326)

4.       Please clarify the sentence “The is because higher sugars consumption stimulates inflammatory immune responses and leads to dysbiosis, which negatively affects sleep” on page 5, lines 175-177 and “and et.al.” on page 8, line 339.

5.       Please format the references according to the journal's instructions

Author Response

We gratefully thank the editor and the reviewer for their time spend making their constructive remarks and suggestions for submitting to International Journal of Molecular Sciences with ID 3305376. Below the comments of the reviewer are responses point by point and the revisions are indicated.

Editors Comments:

  1.  For bacteria please italicize family, genus, species, and variety or subspecies, when used in the singular through the manuscript (example, Bacteroides, Parabacteroides, and Escherichia on page 3 lines 126, 127; Lactobacillus rhamnosus page 4 line 154).

Responses:

Thanks for your kind remind. We have revised them.

Editors Comments:

  1. Please define allabbreviation at their first occurrence in the text (example, HPA on page 2, line 72; FOS, GOS page 5, line 207)

Responses:

Thanks for your kind suggestion. We have added the full name for FOS and GOS when they were used at the first time.

Editors Comments:

  1. After an abbreviation was defined useonly the abbreviation in the text (example, Fecal Microbiota Transplantation on page 8, line 326)

Responses:

Thanks for your kind remind. We have revised with the abbreviations.

Editors Comments:

  1. Please clarify the sentence “The is because higher sugars consumption stimulates inflammatory immune responses and leads to dysbiosis, which negatively affects sleep” on page 5, lines 175-177 and “and et.al.” on page 8, line 339.

Responses:

Thanks for your constructive suggestion.

A high glycemic index and glycemic load diet are associated with an increased risks of insomnia and poor sleep quality. This is because prolonged high sugar intake can stimulate the immune system and induce excessive inflammatory responses, leading to fragmented sleep.

Editors Comments:

  1. Please format the references according to the journal's instructions

Responses:

Thanks for your constructive comments. We have formatted the references according to the journal's instructions

Reviewer 2 Report

Comments and Suggestions for Authors

This is an interesting review, the authors trying to collect all the existing knowledge on the use of microbiome-changing efforts in order to improve insomnia.  Last years it is well documented that microbiome, as a player of the gut-brain axis, is strongly implicated in sleep disorders; and there are many studies dealing with psychobiotics, which beneficially affect insomnia in general, as well as more detailed expression of sleep disorders that is wake up early in the morning or in the middle of the night oe delay to sleep.

Authors try in only four pages to analyzed the influence of prebiotics, probiotics, postbiotics, synbiotics and fecal transplantation in insomnia; and they totally ingnore the new term 'psychobiotics', neither in the text nor in the references.

1. I would like to improve their text, focusing on psychobiotics.

2. Although there is no discussion, I would like to read a comment on fecal transplantation for insomnia, when there are alternatives easily taken. This is a review article, not simply a catalog of related papers.

3. I would like the authors to add tables for each category analyzed, which will include the name of regime used, the type of study [clinical/experimental], the animal used in the case of experiment], the way of documentation of the beneficial effects, and the no of reference

Why the literature is written as "first author and et al", only? [ref no 75 has missed data]

I prefer to change ref 85 of Yesilyurt n et al [not being PubMed but only Scopus cited] OR to keep it but use - along with ref 84 written by Vinterola - the initial statement of postbiotics, that is the consensus statement of the ISAPP [Salminen S, Collado MC, Endo A, Hill C, Lebeer S, Quigley EMM, Sanders ME, Shamir R, Swann JR, Szajewska H, Vinderola G. The International Scientific Association of Probiotics and Prebiotics (ISAPP) consensus statement on the definition and scope of postbiotics. Nat Rev Gastroenterol Hepatol. 2021 Sep;18(9):649-667. doi: 10.1038/s41575-021-00440-6]. 

Author Response

We gratefully thank the editor and the reviewer for their time spend making their constructive remarks and suggestions for submitting to International Journal of Molecular Sciences with ID 3305376. Below the comments of the reviewer are responses point by point and the revisions are indicated.

Editors Comments:

  1. I would like to improve their text, focusing onpsychobiotics.

Responses:

Thanks for your insightful suggestions. Insomnia is indeed classified as a mental health disorder, and there is substantial evidence suggesting that psychobiotics, which influence mental health by modulating the gut microbiome, have the potential to improve sleep. To enhance the content, we have revised the probiotics chapter to place greater emphasis on psychobiotics, as detailed below. The revised sections have been highlighted in blue for your convenience:

Probiotics are live microorganisms that support host health by promoting a balanced gut microbiota composition [73]. Probiotics can remodel the gut microbiota  by competing nutrients with harmful bacterias, producing beneficial metabolites, and modulating the host's immune response [74]. With the growing understanding of the microbiome's role in sleep, probiotics supplementation has been emerged as an effective therapy for alleviating insomnia (Table 1). Rodent studies have demonstrated the efficacy of probiotics supplement in sleep improvement. Mice administered milk fermented with the probiotics Lactobacillus brevis DL1-11 exhibited shorter sleep latency and longer sleep duration compared to those receiving a placebo [75]. Because DL1-11 can produce large amount of GABA, which can promote relaxation and enhance sleep. Another study reported that Lactobacillus fermentum PS150TM effectively promoted sleep in normal mice and alleviated insomnia symptoms in caffeine-induced insomnia mice by increasing the expression of adenosine A1 receptors in the hypothalamus [76]. The same team also discovered that PS150TM could restore NREM sleep in mice with sleep disturbance caused by the first night effect (FNE), likely due to its ability to remodeling gut microbiota composition [77]. The FNE refers to a phenomenon observed in sleep studies where individuals exhibit heightened alertness and anxiety when exposed to an unfamiliar environment, often resulting in acute insomnia [78]. There is no doubt that sleep disorders are closely linked to levels of stress and anxiety. Moreover, many probiotics have been shown to concurrently improve both sleep quality and mental health. PS150TM has been shown to reduce depression and anxiety in rats with chronic mild stress by altering serotonergic pathway [79]. This highlights the potential of psychobiotics—probiotics yielding mental health benefits when administered in adequate amounts—in enhancing sleep [80].

Numerous human trials demonstrated promise of a psychobiotics in enhancing sleep. Bifidobacterium supplement has garnered attention for its potential to improve sleep quality, particularly in individuals experiencing stress. Bifidobacterium breve CCFM1025 improved sleep and alleviated stress levels in patients diagnosed with stress-induced insomnia by suppressing HPA axis activity [81]. Bifidobacterium longum 1714 treatment significantly improved sleep quality and reduced daytime dysfunction in healthy adults experiencing sleep impairment due to the examination stress [82]. However, the EEG monitoring results in this study did not exhibit significant changes. Complementary research on 1714 indicated no significant change in PSQI scores but noted an increase in overall sleep duration [83]. Bifidobacterium adolescentis SBT2786 could prolong total sleep duration by increasing REM sleep time, though it did not enhance subjective sleep quality [84]. Interestingly, researchers found that participants with high stress levels experienced greater benefits from SBT2786, showing significant improvements in sleep duration and reduced daytime fatigue. Although there is  insufficient evidence to classify SBT2786 as a psychobiotic, the subgroup analysis of individuals under high stress indicates its potential for mental health. 

Many Lactobacillus strains are also potential psychobiotics which can promote sleep. Lactobacillus plantarum PS128 significantly enhanced deep sleep, decreased fatigue and depressive symptoms in self-reported insomniacs [85]. Rodent study showed PS128 could increase serotonin and dopamine in the striatum, which may be the underlying mechanism of the beneficial effects [86]. Lactobacillus gasseri CP2305 has also been shown to improve intestinal microbiota composition and reduce salivary cortisol concentrations, thereby alleviating stress and enhancing sleep [87]. Lactobacillus casei Shirota milk helped individuals facing academic examination stress by prolonging sleep duration and reducing feelings of sleepiness upon awakening [88]. The EEG monitoring further revealed the interactions among sleep, stress, and probiotics. As the exam approached, participants experienced increased difficulties with sleep, including prolonged sleep latency and reduced NREM stage 3 sleep duration. However, the intervention with Shirota significantly mitigated these adverse effects. Shirota has also been reported to effectively improving daytime performance in individuals experiencing sleep disorders [89]. Lactococcus lactis subsp. cremoris YRC3780 also demonstrates potential for improving sleep and mitigating stress, as evidenced by a double-blind, placebo-controlled clinical trial conducted in Japan [90]. The study reported significant improvements in subjective sleep quality and mental health among participants receiving the YRC3780 intervention. Although no substantial differences were observed in EEG recordings, lower morning cortisol levels were noted in the YRC3780 group. This suggests that the objective enhancement in sleep may be associated with a reduced stress response within the HPA axis.

Editors Comments:

  1. Although there is no discussion, I would like to read a comment on fecal transplantation for insomnia, when there are alternatives easily taken. This is a review article, not simply a catalog of related papers.

Responses:

Thanks for your constructive comments. We think currently FMT may not the best therapy for insomnia if there are alternatives easily taken. Given our limited understanding of the gut microbiome, we do not fully understand the roles that various microbes play in human health and disease. In this context, microbiota derived from healthy donors are more likely to yield overall improvements for patients. As indicated by the studies referenced in the manuscript, FMT research has investigated insomnia as a comorbidity associated with other health conditions, suggesting that enhancements in sleep may also be attributed to overall improvements in gut microbiota. Due to the uncertainty surrounding the mechanisms underlying any observed benefits, the efficacy of FMT warrants further investigation, particularly regarding complex issues such as sleep. Moreover, the safety of FMT must be carefully considered. Currently, safety assessments for FMT samples are based on the identification of known pathogens and the comprehensive health status of the donors. However, it remains difficult to determine whether the donor's microbiota may contain opportunistic harmful bacteria.

Editors Comments:

  1. I would like the authors toadd tablesfor each category analyzed, which will include the name of regime used, the type of study [clinical/experimental], the animal used in the case of experiment], the way of documentation of the beneficial effects, and the no of reference

Responses:

Thank you very much for your kind suggestion. We have added tables for the paper.

Round 2

Reviewer 2 Report

Comments and Suggestions for Authors

thanks for covering all my comments